# Personalized Management of Hydroxyquinoline Hypersensitivity in Pessary Care: A Case-Based Approach to Tailored Treatment

**DOI:** 10.3390/reports8030145

**Published:** 2025-08-15

**Authors:** Nadege Assassi, Lindsay Robinson, Cathy Zhang, Jill Maura Rabin

**Affiliations:** Department of Obstetrics and Gynecology, Long Island Jewish Medical Center, 270-05 76th Avenue, New Hyde Park, NY 11040, USA; lindsayrobinsonm@gmail.com (L.R.); czhang20@northwell.edu (C.Z.); jrabin@northwell.edu (J.M.R.)

**Keywords:** hydroxyquinoline, pelvic organ prolapse, pessary, dermatitis, allergy, case report

## Abstract

**Background and Clinical Significance**: Many women use pessaries to manage their symptoms of pelvic organ prolapse. Hydroxyquinoline is the active ingredient in gels and ointments that are often used to lubricate a pessary prior to vaginal insertion and to provide antimicrobial effects while the pessary is in situ. **Case Presentation**: A 74-year-old woman with multiple medication allergies develops vulvovaginal erythema and pruritus after increasing vaginal Trimo-San application frequency for pessary care and maintenance. These symptoms are deemed to be consistent with an allergic reaction to hydroxyquinoline, the active ingredient in Trimo-San. **Conclusions**: This report highlights the importance of personalized treatment in pessary management. It also demonstrates how personalized medicine can optimize outcomes and improve treatment adherence among individuals with complex medical histories.

## 1. Introduction and Clinical Significance

Pelvic organ prolapse (POP) is a benign condition in which portions of the vagina, cervix, and uterus descend through their openings in the levator musculature, causing the herniation of these organs into the vagina. In the United States alone, 3–6% of women report symptoms of POP, most notably a vaginal bulge [1]. POP is predominantly due to pelvic muscle weakness or defects within the connective tissue that support each organ. Risk factors for the development of such muscle and connective tissue weakness include multiparity, history of vaginal delivery, elevated BMI, and older age [2,3].

A variety of surgical and nonsurgical interventions are available to manage POP. Method of management is determined primarily based on the patient’s preference and degree of symptom bother, with consideration of any pre-existing conditions, comorbidities, or contraindications to surgery. The least invasive and most commonly used option for management of POP is insertion of a fitted vaginal pessary. Because POP often occurs in postmenopausal women, and therefore is associated with low serum estrogen levels, concomitant vulvovaginal atrophy is common [4]. Consequently, if a pessary is left in situ for extended periods, compression of the friable, thin vulvovaginal tissue can lead to local tissue devascularization and facilitate skin erosion or abrasions. For this reason, vaginal pessaries are often removed routinely at varying time intervals, based on the type of pessary, and coated with a lubricating gel, cream, or ointment prior to insertion to protect the vaginal tissue from irritation [5]. Trimo-San, which contains the active ingredient hydroxyquinoline, is an example of such a gel. Trimo-San is an acidic lubricant that, when applied topically, keeps the pH of the vagina low to promote growth of lactobacillus and other microbes of the normal vaginal microbiome, while inhibiting growth of infectious microbes [6,7,8]. This case report describes a dermopathy suspected to be an allergic reaction to vulvovaginal topical hydroxyquinoline use in the setting of routine pessary care.

## 2. Case Presentation

A 74-year-old gravida 2 para 2 woman, with multiple antibiotic and antihypertensive medication allergies (e.g., penicillin, sulfa, and captopril) and a medical history of hypertension, hyperlipidemia, hypothyroidism, and renal calculi, presents to the urogynecology office with a chief concern of vulvovaginal itching and redness for 3 days. 

The patient has a history of stage IV POP diagnosed 3 years ago. She initially attempted to manage her symptoms of POP with pelvic floor physical therapy. However, her prolapse continued to progress despite intervention. She was counseled on surgical and nonsurgical management alternatives. She elected to manage her symptoms with a pessary, and she has since been using a 2.75 in ring pessary. When initially fitted for the pessary, the patient was instructed to remove it weekly to clean it, and to re-insert the pessary using lubricating Trimo-San gel. While she had been compliant thus far with daily perineal hygiene and weekly pessary removal for routine cleaning as instructed, the patient admitted to recently accidentally leaving the pessary in situ for 11 days when she went on vacation. Upon pessary removal after this extended period, the patient noticed new onset vulvovaginal redness, itching, and vaginal spotting originating from the prolapsing vaginal walls. To attempt to avoid further irritation to the vulvovaginal tissue, the patient began applying Trimo-San gel daily instead of weekly and used larger quantities with each application. However, despite more frequent application and daily pessary removal, there was no improvement in the vulvar pruritus and erythema. Therefore, the patient presented to the urogynecology office for further evaluation.

On evaluation in the office, the patient reported spontaneous resolution of vaginal spotting. She endorsed symptoms of a vaginal bulge, consistent with her diagnosis of POP. She denied vaginal discharge, pelvic pain, urinary or fecal incontinence, urinary urgency, frequency, dysuria, or obstructive voiding symptoms. Physical exam was significant for stage IV POP and severe vulvovaginal atrophy. A 3 × 2 mm anterior cervical ulceration without erosion was noted, but there were otherwise no vaginal or cervical lesions appreciated (Figure 1). 

### 2.1. Differential Diagnosis

Vulvovaginal erythema and pruritus are associated with a multitude of gynecologic and urologic etiologies. Common causes of an acute presentation of these symptoms include infection. Therefore, it is important to evaluate for the presence of bacterial, viral, or fungal infection. The patient underwent a urinalysis with reflex urine culture and an Affirm test to assess for the presence of Gardnerella, Trichomonas, and Candida infection. Urine sampling yielded no microbial growth. Affirm testing yielded Gardnerella growth but demonstrated no growth of Trichomonas or Candida. The decision was made to defer treatment with metronidazole, as the patient did not have any other typical symptoms of Gardnerella infection (e.g., foul-smelling vaginal discharge). 

Allergic reactions could also present with acute cutaneous changes as seen in this patient. An allergic reaction to the patient’s pessary was considered. Pessaries are most commonly manufactured using silicone. A review of the literature suggests that allergic reactions to medical implants and devices made of silicone are rare. There is no level 1 evidence to indicate that silicone is a potent allergen associated with the development of contact dermatitis symptoms, as such an allergic reaction has only been described in small case reports, particularly with regard to the use of silicone ventriculoperitoneal shunts and breast implants [9,10]. To date, there is no report in the literature of vulvovaginal contact dermatitis developing in the setting of silicone pessary use. This patient was known to have many pre-existing allergies to antimicrobial and antihypertensive medications. Given this history and the acute development of localized vulvovaginal erythema and pruritus after the patient increased the dose of Trimo-San applied to the vulvovaginal tissue, the most probable etiology of the patient’s symptoms was deemed to be an allergic reaction to hydroxyquinoline, the active ingredient in Trimo-San. The diagnosis of an allergic reaction to hydroxyquinoline was further supported by resolution of the patient’s symptoms shortly after she discontinued Trimo-San use despite continued pessary use for management of her POP symptoms. 

### 2.2. Treatment and Outcome

The patient was instructed to remove the pessary and start using triple paste barrier cream daily in place of Trimo-San. Ulceration of the cervix improved within 48 h of discontinuing Trimo-San application. Within 5 days of discontinuing Trimo-San, mucosal erythema had improved. Within 1 week, the patient noted improvement in vulvovaginal symptoms, and the mucosa was restored to a grossly normal appearance (Figure 2). The patient has started using a pessary again and removes it weekly for routine cleaning. She has been using vaginal estrogen cream in place of Trimo-San for pessary insertion with no consequent vulvovaginal side effects (Figure 3). Vaginal estrogen cream was selected for this patient based on her vulvovaginal atrophy and history of medication allergies that favored topical rather than systemic therapy. This approach offered a sustainable, long-term solution to address the patient’s underlying tissue vulnerability while allowing for continued pessary use. Following the discontinuation of Trimo-San, the patient expressed great appreciation and relief with her clinical improvement. 

## 3. Discussion

This case report highlights several clinical considerations in pessary management, medication hypersensitivity, and personalized medicine in urogynecologic care. POP is a common and debilitating condition that affects many women worldwide. Treatment of POP can improve patients’ quality of life by improving bulge symptoms, as well as pelvic pain and urinary and fecal incontinence that often affect this patient population [11,12]. While many women with symptomatic POP ultimately choose to undergo surgical correction, some are not eligible for surgical management, and others prefer non-surgical alternatives. Vaginal pessaries are a popular and effective non-surgical intervention for the management of POP in this patient population.

There are two main categories of pessaries: support and space-occupying pessaries. Support pessaries, such as ring pessaries, are easy to fit, insert, and remove, and can be left in situ during intercourse. Space-occupying pessaries provide more apical support than support pessaries, but they must be removed for intercourse. Space-occupying pessaries are more commonly used in patients with stage IV POP and a widened genital hiatus. Despite the large selection of pessary options available to patients in recent years, long-term compliance with pessary insertion and cleaning remains variable. This is, in part, attributed to inherent side effects associated with different pessaries that discourage continued pessary use. Examples of such side effects include malodorous vaginal discharge and vaginal wall irritation or erosion from pessary insertion [11,13].

Pessaries are flexible, yet remain largely static within the vagina, to continue providing adequate pelvic floor support despite movement, activity, or a change in the configuration of the vagina or extent of POP. Thus, unsurprisingly, one of the most commonly cited reasons for pessary discontinuation is pelvic pain or discomfort [13,14]. To minimize adverse effects and maximize compliance, providers often prescribe lubricants to apply to the vaginal walls and the pessary to facilitate pessary placement and improve comfort with the pessary in place. One commonly used lubricant is Trimo-San, an acidic gel, which, when applied, lowers the pH of the vagina, thus preventing microbial overgrowth and disruption of the vaginal microbiome [6,15]. Hydroxyquinoline, the active ingredient in Trimo-San, also exhibits broad-spectrum antimicrobial activity against Gram-positive and Gram-negative bacteria, fungi, and certain viruses through metal chelation mechanisms that disrupt essential cellular processes [8]. Hydroxyquinoline also has anti-biofilm properties, inhibiting growth of bacterial colonies that cause symptomatic vaginal infections [7]. However, these same therapeutic properties that make hydroxyquinoline valuable in pessary care can also contribute to hypersensitivity reactions, particularly with excessive use or use in individuals with compromised skin barriers, as demonstrated in dermatological literature [16]. Informed by an individualized risk assessment, the known benefits of topical hydroxyquinoline in pessary care must be weighed against its potential to cause hypersensitivity or allergic reactions before recommending hydroxyquinoline-based topical products to patients with POP.

Though the discontinuation of pessary use can often be attributed to adverse effects from the pessary itself (e.g., pessary shape or size), this case report describes an allergic reaction to the use of Trimo-San gel with pessary insertion that, without treatment, can discourage continued use of a pessary to manage POP symptoms. A comprehensive literature review reveals remarkably few reports documenting allergic reactions to hydroxyquinoline-based products, despite their widespread clinical use. While hydroxyquinoline compounds have been utilized in various medical applications for decades, documented cases of hypersensitivity reactions are limited to isolated case reports, primarily involving systemic administration of related compounds such as clioquinol [17]. The scarcity of published data on adverse reactions to hydroxyquinoline may reflect under-recognition, underreporting, or genuine rarity of such reactions. This report documents a localized, rather than systemic, allergic cutaneous reaction following vulvovaginal application of hydroxyquinoline-based topical lubricant for pessary care, describing an adverse effect that has not yet been reported in this clinical setting. 

Recognition of hydroxyquinoline hypersensitivity is clinically important for several reasons. Since hydroxyquinoline-based products are commonly used in pessary care, allergic reactions can mimic urogenital infections, potentially leading to inappropriate antimicrobial therapy if not correctly and promptly identified. Healthcare providers should maintain high suspicion for allergic reactions in patients with medication allergy histories who develop vulvovaginal irritation following pessary insertion, particularly when symptoms occur after increased application frequency or quantity, as demonstrated in this case report.

This case exemplifies both personalized medicine principles and the importance of shared decision-making in pessary care. The patient’s multiple risk factors, including extensive medication hypersensitivity history, severe vulvovaginal atrophy, and cardiovascular comorbidities, contributed to her susceptibility to hydroxyquinoline reaction. Our tailored approach led to successful substitution with vaginal estrogen cream, which simultaneously addressed her underlying tissue atrophy while resolving her bothersome vulvar cutaneous symptoms by eliminating the identified allergen.

Effective pessary management requires a collaborative partnership between patients and providers, emphasizing comprehensive education on proper application techniques, potential adverse reactions, and recognition of symptoms of such adverse reactions. Patients with complex medical histories benefit from enhanced counseling that includes discussion of alternative topical lubricating and antimicrobial products for pessary maintenance and individualized strategies to monitor for hypersensitivity reactions and other sequelae from hydroxyquinoline use. Educational materials should address proper pessary hygiene and maintenance, differentiation between allergic and infectious symptoms, appropriate method of topical lubricant use, and clear guidance on when to seek medical evaluation for untoward effects. In this case, the patient’s active participation in symptom pattern recognition and her prompt adherence to a modified pessary care routine were essential to achieving successful outcomes.

The development of the following Management Protocol for Suspected Hydroxyquinoline Hypersensitivity has been informed by the principles of personalized medicine that were incorporated in this patient’s care:

Immediate Actions: discontinue hydroxyquinoline-containing product, remove pessary, document symptoms.

Diagnostic Workup: rule out infection, assess for mechanical causes of symptoms, review allergy history. 

Initial Treatment: apply barrier cream to affected area, pelvic rest.

Alternative Therapy: choose a substitute topical lubricant based on individual risk factors and tissue characteristics (e.g., atrophy).

Resumption of Pessary Use: confirm improvement in tissue integrity and patient’s symptoms, restart pessary use with alternative topical lubricant to facilitate pessary insertion, counsel patient on recommended dose and frequency of lubricant use.

Long-term Maintenance Treatment: encourage individualized regular monitoring of symptoms with lubricant use, prevent untoward side effects by stratifying patients based on risk factors for developing side effects from topical hydroxyquinoline product use.

This evidence-based approach can serve as a template for clinicians managing patients with similar symptoms in urogynecologic practice. Applying this individualized approach to pessary maintenance and care may help prevent the occurrence of adverse reactions through proactive risk assessment and comprehensive patient education, thus improving adherence to pessary use for non-surgical management of POP.

A Cochrane review published in 2020 demonstrated that patients who use pessaries to manage POP experience more adverse effects than patients who rely instead on other nonsurgical interventions, such as pelvic floor physical therapy [11]. Such adverse effects include vaginal discharge, urinary incontinence, and irritation of vaginal walls [2,3]. However, there are no publications that have analyzed the occurrence of adverse effects based on the type of lubricant used for pessary insertion and routine care. This case highlights the importance of considering personal allergy history, individual tissue quality, and patient-specific behaviors when selecting pessary management strategies, particularly for patients with complex medical histories or medication sensitivities. Though we only present the management and treatment response of one patient, the findings presented in this case report indicate the need for high-quality studies to investigate the prevalence of adverse reactions to hydroxyquinoline in pessary users and to develop evidence-based recommendations for optimal frequency of routine pessary removal, cleaning, and re-insertion.

## 4. Conclusions

This case documents a case of localized allergic contact dermatitis triggered by hydroxyquinoline-based products in pessary care, demonstrating that this widely used product can cause serious adverse reactions, especially in patients with known sensitivities to other medications. This case illustrates how personalized medicine elements such as comprehensive risk assessment, tailored therapeutic selection, and individualized monitoring can optimize outcomes, especially when standard approaches have already failed. Clinicians are encouraged to refer to a patient’s medication and allergy history to inform their recommendations for pessary management. This case report also calls on clinicians to monitor for hydroxyquinoline sensitivity periodically in patients using pessaries to manage POP symptoms, and to consider alternatives to hydroxyquinoline-based lubricants, such as vaginal estrogen cream, especially for patients with atrophic vulvovaginal tissue or medication sensitivities. Patient education on the optimal dose and application frequency of hydroxyquinoline-based lubricants is essential, as dose-dependent reactions may occur, even after prolonged, uneventful use. This case highlights a need for further research on the prevalence of hydroxyquinoline sensitivity and evidence-based guidelines for personalized lubricant selection in pessary management. 

## Figures and Tables

**Figure 1 reports-08-00145-f001:**
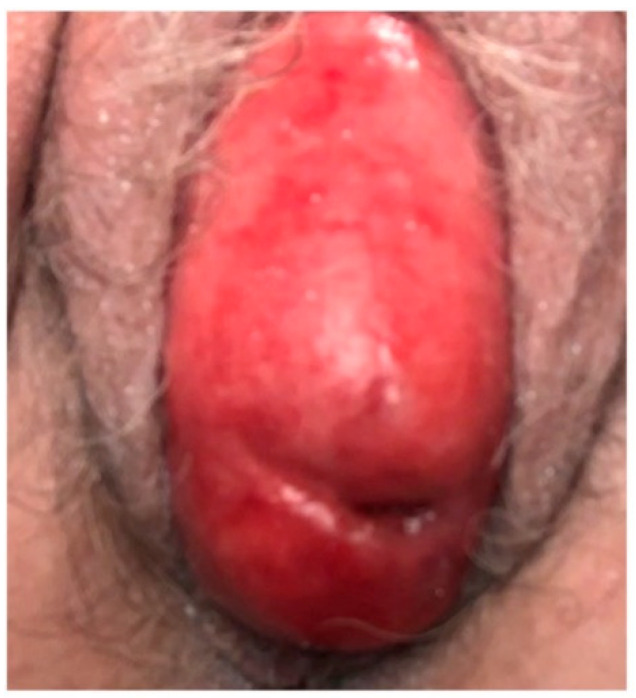
Vaginal epithelium at initial office evaluation.

**Figure 2 reports-08-00145-f002:**
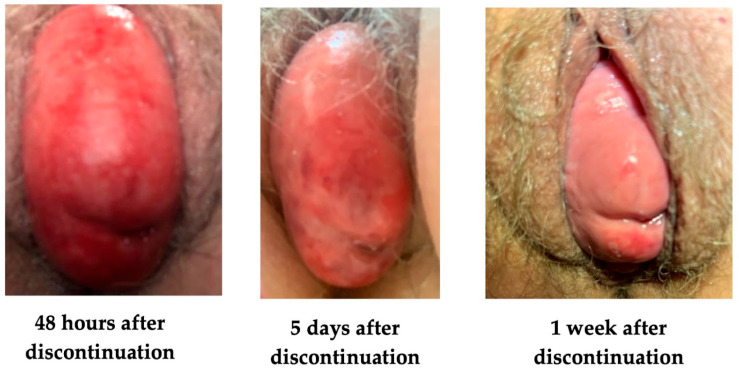
Vaginal epithelium after Trimo-San discontinuation.

**Figure 3 reports-08-00145-f003:**
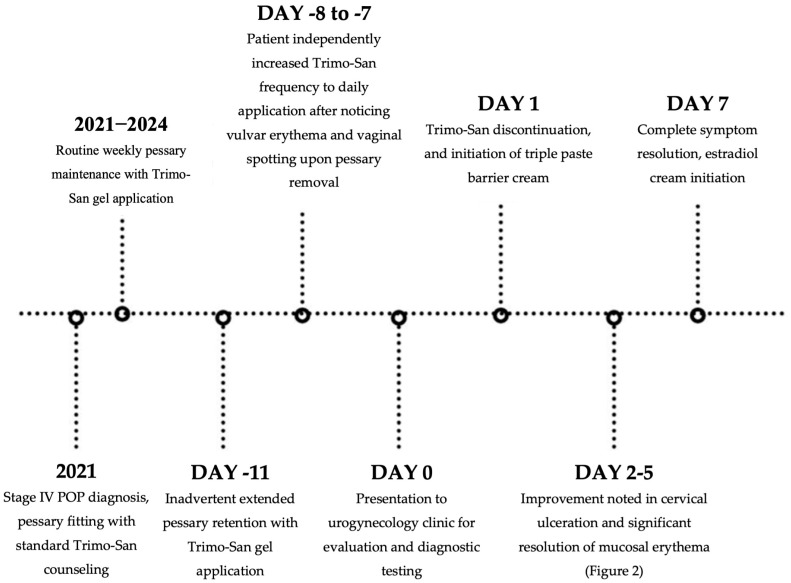
Case Timeline.

## Data Availability

The original data presented in the study are included in the article. Further inquiries can be directed to the corresponding author.

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
