# Peer review of "Personalized Management of Hydroxyquinoline Hypersensitivity in Pessary Care: A Case-Based Approach to Tailored Treatment"

_reports, 2025, doi:10.3390/reports8030145_

Round 1
Reviewer 1 Report
Comments and Suggestions for Authors
This article presents a valuable case report on hydroxyquinoline hypersensitivity in pessary care, highlighting the importance of personalized management. The narrative is clear, and the clinical reasoning is sound. To further enhance its quality and impact for publication, I offer the following critical comments:
Abstract:
Conciseness: While informative, the abstract could be slightly more concise. For instance, the phrase "highlighting the importance of personalized risk assessment and individualized treatment selection in pessary management" is a bit long. It could be streamlined without losing meaning.
Introduction:
Citation Consistency: Please review the citation numbering. The first citation for POP definition is , while the subsequent citations are and . This unusual jump (starting with then going to ``) suggests a potential renumbering issue or a typo. Ensure the citations are sequential and correct according to your chosen referencing style.
Case Presentation:
Missing Figures and Figure Captions: This is a critical issue for a case report. Figures 1, 2, 3, and 4 are referenced in the text, but the figures themselves are absent. Furthermore, the captions for these figures are either missing or incorrectly formatted (e.g., "Figure 1Differential Diagnosis:."). For a case report, visual documentation is often crucial for illustrating clinical findings and supporting the narrative.
Recommendation: Ensure all referenced figures are included with clear, descriptive captions that are properly formatted and distinct from the main text.
Flow of Diagnostic Process: The description of the differential diagnosis and the process of ruling out infection is well-structured and logical. The justification for deferring metronidazole is also clear.
Support for Diagnosis: The reasoning for concluding an allergic reaction to hydroxyquinoline (patient's allergy history, acute onset after increased dose/frequency, and resolution upon discontinuation) is well-supported by the clinical details provided.
Discussion:
Novelty Claim: The statement "this is one of a limited set of reports describing an allergic reaction to hydroxyquinoline-based products, and the first report of a localized allergic reaction after vulvovaginal application of a topical formulation of a hydroxyquinoline-based product " is a strong claim of novelty. Please ensure that citation `` directly and robustly supports this specific claim, particularly the "first report of a localized allergic reaction after vulvovaginal application of a topical formulation."
Citation Consistency/Accuracy (again): In the discussion, citation is used again for a Cochrane review on adverse effects of pessaries. This same citation was used in the introduction for risk factors of POP. While it's possible for one source to cover both, it's unusual for a single general citation to refer to a specific Cochrane review. Please verify that `` indeed refers to a Cochrane review that supports this specific point, or provide a separate, more precise citation if needed.
Dose-Dependent Effect: The emphasis on the potential dose-dependent nature of the adverse reaction and the importance of patient counseling is a valuable clinical takeaway from this case.
Personalized Medicine Theme: The discussion effectively links the case to the principles of personalized medicine, demonstrating how the patient's complex history necessitated a tailored approach.
Alternative Treatment (Estradiol): The rationale for using topical estradiol as an alternative lubricant and treatment for vulvovaginal atrophy is well-explained and clinically relevant.
Research Gap and Future Directions: The identification of the research gap regarding adverse effects of different lubricants and the call for high-quality studies on hydroxyquinoline reactions and optimal pessary care frequency are excellent suggestions for future research.
Author Response
Comment 1: This article presents a valuable case report on hydroxyquinoline hypersensitivity in pessary care, highlighting the importance of personalized management. The narrative is clear, and the clinical reasoning is sound. To further enhance its quality and impact for publication, I offer the following critical comments:
Abstract:
Conciseness: While informative, the abstract could be slightly more concise. For instance, the phrase "highlighting the importance of personalized risk assessment and individualized treatment selection in pessary management" is a bit long. It could be streamlined without losing meaning.
Response 1: On lines 10-17, the abstract was edited for brevity. Sentences were made shorter to improve readability.
Comment 2: Introduction:
Citation Consistency: Please review the citation numbering. The first citation for POP definition is , while the subsequent citations are and . This unusual jump (starting with then going to ``) suggests a potential renumbering issue or a typo. Ensure the citations are sequential and correct according to your chosen referencing style.
Response 2: Citations are numbered in alphabetical order in keeping with APA citation formatting. Therefore, citation numbers do not appear in chronological order throughout the manuscript. Citation numbering has been restructured to accommodate additional sources, and the in-text citations have been re-numbered within the manuscript to reflect the addition of new sources. If citation structure is required to be sequential rather than alphabetical, please do not hesitate to let us know and we will promptly make the adjustments to our in-text numbering system.
Comment 3: Case Presentation:
Missing Figures and Figure Captions: This is a critical issue for a case report. Figures 1, 2, 3, and 4 are referenced in the text, but the figures themselves are absent. Furthermore, the captions for these figures are either missing or incorrectly formatted (e.g., "Figure 1Differential Diagnosis:."). For a case report, visual documentation is often crucial for illustrating clinical findings and supporting the narrative. Recommendation: Ensure all referenced figures are included with clear, descriptive captions that are properly formatted and distinct from the main text.
Response 3: We appreciate this being brought to our attention. Lines 69-85 and lines 129-131 now contain all figures with corresponding captions/descriptions of each image.
Comment 4: Flow of Diagnostic Process: The description of the differential diagnosis and the process of ruling out infection is well-structured and logical. The justification for deferring metronidazole is also clear. Support for Diagnosis: The reasoning for concluding an allergic reaction to hydroxyquinoline (patient's allergy history, acute onset after increased dose/frequency, and resolution upon discontinuation) is well-supported by the clinical details provided.
Discussion: Novelty Claim: The statement "this is one of a limited set of reports describing an allergic reaction to hydroxyquinoline-based products, and the first report of a localized allergic reaction after vulvovaginal application of a topical formulation of a hydroxyquinoline-based product " is a strong claim of novelty. Please ensure that citation `` directly and robustly supports this specific claim, particularly the "first report of a localized allergic reaction after vulvovaginal application of a topical formulation."
Response 4: On lines 173-181, the methodology of our literature review has been described in further detail to support our understanding that there is limited research published about the prevalence of allergic reactions to hydroxyquinoline products, and that this case report will fill a gap in the literature regarding how to care for patients with prolapse via non-surgical approaches. On lines 85-97, we now also include cited descriptions of contact dermatitis reactions with use of silicone medical devices other than pessaries to highlight that we considered the patient’s symptoms could be related to the use of a pessary itself, rather than the use of a lubricant for pessary insertion.
Comment 5: Citation Consistency/Accuracy (again): In the discussion, citation is used again for a Cochrane review on adverse effects of pessaries. This same citation was used in the introduction for risk factors of POP. While it's possible for one source to cover both, it's unusual for a single general citation to refer to a specific Cochrane review. Please verify that `` indeed refers to a Cochrane review that supports this specific point, or provide a separate, more precise citation if needed.
Dose-Dependent Effect: The emphasis on the potential dose-dependent nature of the adverse reaction and the importance of patient counseling is a valuable clinical takeaway from this case. Personalized Medicine Theme: The discussion effectively links the case to the principles of personalized medicine, demonstrating how the patient's complex history necessitated a tailored approach. Alternative Treatment (Estradiol): The rationale for using topical estradiol as an alternative lubricant and treatment for vulvovaginal atrophy is well-explained and clinically relevant. Research Gap and Future Directions: The identification of the research gap regarding adverse effects of different lubricants and the call for high-quality studies on hydroxyquinoline reactions and optimal pessary care frequency are excellent suggestions for future research.
Response 5: We sincerely appreciate the positive feedback. On lines 231-234, while the Cochrane review remains cited, additional sources have now been cited to more specifically describe the adverse effects associated with pessary use. Risk factors associated with POP as outlined in the introduction are now also supported by more precise citations.

Reviewer 2 Report
Comments and Suggestions for Authors
The case report titled "Personalized Management of Hydroxyquinoline Hypersensitivity in Pessary Care: A Case-Based Approach to Tailored Treatment" presents a rare allergic reaction to hydroxyquinoline in a 74-year-old woman using a pessary for pelvic organ prolapse (POP). The report highlights the importance of individualized care in urogynecology, particularly for patients with complex medical histories. While the case is well-documented and clinically relevant, the discussion could be strengthened by broader literature integration and clearer management recommendations.
There are undoubtful strengths of the investigation: addresses a rare but important adverse effect of hydroxyquinoline, a common pessary lubricant, emphasizing personalized medicine in POP management. Thorough history, physical exam, and diagnostic workup (e.g., Affirm testing) rule out infections and support the allergy diagnosis. Highlights the need for patient counseling on pessary care and alternative lubricants (e.g., topical estradiol).
Nevertheless there is a gap to improve some points: limited discussion of prior hydroxyquinoline allergy cases (only one cited). Expand with examples from dermatology or other specialties; propose a step-by-step approach for managing similar cases (e.g., allergy testing, alternative lubricants); patient follow-up: describe long-term outcomes after switching to estradiol (e.g., symptom resolution, pessary compliance).
Moreover, I would like to make some specific comments:
Introduction (Section 1): clarify hydroxyquinoline’s mechanism of action (antimicrobial vs. lubricating) and prevalence in pessary care. Cite studies on pessary-related adverse effects (e.g., Bugge et al., Cochrane 2020) to contextualize the rarity of this reaction.
Figure 1: Include clinical images (with consent) to illustrate vulvovaginal erythema. Strengthen the differential diagnosis by comparing hydroxyquinoline allergy to contact dermatitis from other pessary materials (e.g., silicone).
Discussion (Section 3): Contrast this case with prior reports of systemic hydroxyquinoline allergies (e.g., Janier & Vignon, 1995). Discuss potential cross-reactivity with other quinoline derivatives or preservatives in vaginal products. Emphasize shared decision-making in pessary care (e.g., patient education materials).
Author Response
Comment 1: The case report titled "Personalized Management of Hydroxyquinoline Hypersensitivity in Pessary Care: A Case-Based Approach to Tailored Treatment" presents a rare allergic reaction to hydroxyquinoline in a 74-year-old woman using a pessary for pelvic organ prolapse (POP). The report highlights the importance of individualized care in urogynecology, particularly for patients with complex medical histories. While the case is well-documented and clinically relevant, the discussion could be strengthened by broader literature integration and clearer management recommendations.
There are undoubtful strengths of the investigation: addresses a rare but important adverse effect of hydroxyquinoline, a common pessary lubricant, emphasizing personalized medicine in POP management. Thorough history, physical exam, and diagnostic workup (e.g., Affirm testing) rule out infections and support the allergy diagnosis. Highlights the need for patient counseling on pessary care and alternative lubricants (e.g., topical estradiol).
Response 1: We agree with this comment, and have made adjustments to improve the strength of our discussion section. Found on lines 208-229, we have now included a step-by-step protocol to describe our management recommendation as to how providers should address a concern for hydroxyquinoline hypersensitivity. Additional sources (reference numbers 2, 3, 4, 5, 6, 8, 11, 13, 14, 15, 16, and 17 on the references section) have also been included to broaden the scope of literature we have used to support the individualized evaluation of our patient, and to support our rationale for our diagnosis of a contact dermatitis from Trimo-San use.
Comment 2: Nevertheless there is a gap to improve some points: limited discussion of prior hydroxyquinoline allergy cases (only one cited). Expand with examples from dermatology or other specialties; propose a step-by-step approach for managing similar cases (e.g., allergy testing, alternative lubricants); patient follow-up: describe long-term outcomes after switching to estradiol (e.g., symptom resolution, pessary compliance).
Response 2: We agree with this comment, and have sought to improve the strength of our conclusions with the support of literature from other specialties. On lines 162-229, descriptions of allergic reactions to hydroxyquinoline-related products that are used systemically, such as clioquinol, have been added. A step-by-step approach has also been included to demonstrate how to approach similar cases. On lines 95-101, case reports on silicone medical devices have also been included to demonstrate the consideration that this patient’s symptoms could be due to insertion of the pessary itself. On lines 114-120, we describe the patient’s prompt resolution of symptoms with this approach to treatment since switching to estrogen cream.
Comment 3: Introduction (Section 1): clarify hydroxyquinoline’s mechanism of action (antimicrobial vs. lubricating) and prevalence in pessary care. Cite studies on pessary-related adverse effects (e.g., Bugge et al., Cochrane 2020) to contextualize the rarity of this reaction.
Response 3: On lines 38-41 and lines 157-168, we describe the antimicrobial and lubricating properties of hydroxyquinoline. On lines 95-101, we identify a gap in the literature in describing adverse reactions to silicone pessary use and Trimo-San use. On lines 135-137, we describe common adverse effects to pessary use, and cite 2 sources to support this finding.
Comment 4: Figure 1: Include clinical images (with consent) to illustrate vulvovaginal erythema. Strengthen the differential diagnosis by comparing hydroxyquinoline allergy to contact dermatitis from other pessary materials (e.g., silicone).
Response 4: On lines 69-85 and 129-131, all figures are now included in the manuscript with corresponding captions/descriptions of each image. On lines 95-107, we expand on our process on narrowing down our differential diagnosis by explaining the characteristics of a hydroxyquinoline allergy, and comparing these features to those of a contact dermatitis reaction from silicone, the material with which most modern pessaries are made.
Comment 5: Discussion (Section 3): Contrast this case with prior reports of systemic hydroxyquinoline allergies (e.g., Janier & Vignon, 1995). Discuss potential cross-reactivity with other quinoline derivatives or preservatives in vaginal products. Emphasize shared decision-making in pessary care (e.g., patient education materials).
Response 5: On lines 173-181, systemic hydroxyquinoline allergic reactions are now included in this manuscript, cited, and contrasted from the reaction described in the subject of this case. On lines 190-206, the importance of shared decision making and patient-centered care with patient education on how to optimally manage POP without surgery is now described.

Reviewer 3 Report
Comments and Suggestions for Authors
In this article, authors demonstrated that an allergic patient developed allergy to the hydroxyquinoline used when inserting the pessary. Interesting case. Relevant images.
-Maybe you could say a few more sentences about other actions of hydroxyquinoline, beside lines 133-138: “In an effort to minimize adverse effects and maximize compliance, providers often prescribe lubricants to apply to the vaginal walls and the pessary to facilitate pessary placement and improve comfort with the pessary in place. One commonly used lubricant is Trimo-San, an acidic gel, which, when applied, lowers the pH of the vagina, thus preventing microbial overgrowth and disruption of the vaginal microbiome “ and later line 146 :“ since hydroxyquinoline-based products are commonly used lubricants in this patient population” Please say more about hydroxyquinoline, at least one paragraph.
-In References, out of 11 titles, only 3 are recent, even if they are well chosen titles. Maybe you could add more recent titles.
Author Response
Comment 1: Maybe you could say a few more sentences about other actions of hydroxyquinoline, beside lines 133-138: “In an effort to minimize adverse effects and maximize compliance, providers often prescribe lubricants to apply to the vaginal walls and the pessary to facilitate pessary placement and improve comfort with the pessary in place. One commonly used lubricant is Trimo-San, an acidic gel, which, when applied, lowers the pH of the vagina, thus preventing microbial overgrowth and disruption of the vaginal microbiome “ and later line 146 :“ since hydroxyquinoline-based products are commonly used lubricants in this patient population” Please say more about hydroxyquinoline, at least one paragraph.
Response 1: We agree that more information is warranted regarding hydroxyquinoline's mechanism of action and the relevance of these properties in its use in pessary care. Lines 157-168 now expand further on the mechanism of action of hydroxyquinoline and hydroxyquinoline-based products.
Comment 2: In References, out of 11 titles, only 3 are recent, even if they are well chosen titles. Maybe you could add more recent titles.
Response 2: We agree with this comment. In order to provide the most up-to-date review of relevant data available regarding hydroxyquinoline's use in clinical practice, and in particular, in urogynecologic care, more recent articles have been added such that 6 of the 17 cited references are from 2020 or thereafter. Reference numbers for these sources are 2, 4, 5, 12, 14, and 15 from the references list.

Round 2
Reviewer 1 Report
Comments and Suggestions for Authors
Thank you for your revision.